

# Short-term starvation at low temperature prior to harvest does not impact the health and acute stress response of adult Atlantic salmon

Rune Waagbø[1], Sven Martin Jørgensen[2], Gerrit Timmerhaus[2], Olav Breck[3] and Pål A. Olsvik[1]

[1] National Institute of Nutrition and Seafood Research, Bergen, Norway
[2] Nofima, Ås, Norway
[3] Marine Harvest ASA, Bergen, Norway

## ABSTRACT

A period of starvation is regarded as a sound practice in aquaculture prior to handling, transportation and harvest, to minimise impacts on welfare and ensure proper hygiene after harvest. However, documentation of welfare issues such as stress following starvation and handling in adult Atlantic salmon are lacking. This study aimed to examine gut emptying and potential stress during a two week starvation period, and whether this starvation period changed the tolerance for physical stress. The study confirmed slower emptying of the gut segments at low temperature. Plasma and bile cortisol, and selected clinical analyses were used to characterize potential stress, as well as the response to acute physical crowding stress during the starvation period. Neither the general stress level nor the ability to cope with handling stress was affected by a 14 day starvation period. Down-regulation of selected nutritional related gene markers in liver indicated classical starvation responses, with reduced metabolism and oxidative pressure, and sparing of nutrients. The response to acute handling stress was not affected by two weeks of starvation. There were minor effects of starvation on stress and health markers, as evaluated by plasma lysozyme activity and gene expression of selected inflammation marker proteins in heart and skin tissues.

## INTRODUCTION

Fish are exposed to periods of starvation or restricted feed intakes both in wild and for practical reasons in aquaculture. In these periods, the fish covers the energy requirements at the expense of body stores of nutrients (*Lie & Huse, 1992*). Along with the rapid expanding aquaculture production, increasing concerns on fish welfare and ethically acceptable production practices have called for scientific evaluation of biological and behavioural consequences of feeding and starvation practices (*Lines & Spence, 2012*).

Fasting and feed withdrawal periods prior to transportation and harvest of Atlantic salmon is practiced to obtain complete gut evacuation and a clean digestive tract, to ensure good water quality (e.g., minimize excretion of ammonia) and to reduce metabolic rate,

Corresponding author
Rune Waagbø, rwa@nifes.no, rune.waagbo@nifes.no

physical activity, hierarchy and stress during transportation (*Robb, 2008*; *VKM, 2008*). According to the quality regulations in the Norwegian food legislation, fish should be starved to empty the gut prior to harvest to ensure proper hygiene in further processing. Temperature is the major factor influencing gut evacuation rate (*Usher, Talbot & Eddy, 1991*). They showed that other factors such as feed composition and physical quality may also influence the evacuation time, while fish size seems less important. Large cages in commercial scale aquaculture may need longer periods for harvesting and thus the fish population will be starved for longer periods for practical reasons.

Transportation of live salmon is a stressful event involving handling, crowding and exposure to varying water qualities (*VKM, 2008*). Starvation is practiced also for welfare concerns since there is a general understanding that starved fish are calmer and more tolerant to stress. However, the scientific rationale for this is not substantial and primarily related to reduced metabolic rate being indicative of higher stress tolerance (*Petri, 2003*). Fed fish may be less robust and more susceptible to stressors (e.g., handling) based on the notion that starvation save energy for digestion and metabolic processes, the fish has lower oxygen demands, less waste production and thereby conserving easily available energy for stress coping. There are few studies demonstrating additional beneficial effects on stress tolerance by starving fish for longer periods than three days (*Einen, Waagan & Thomassen, 1998*). *Mørkøre et al. (2008)* concluded that a starvation period of five weeks can apparently improve the resistance to acute stress prior to slaughtering of Atlantic salmon. The pre-harvest starvation period is, however debated from a welfare perspective and standard practices suggest between five and a maximum of 14 days with the priority of a cleared gut (*Robb, 2008*).

Adaptation to starvation includes metabolic adjustments, such as reduced basal metabolic rate, reduced activity in all organs related to exogenous nutrition and swimming activity (*Petri, 2003*). The net result is a reduced spending of stored energy, mainly from intestinal and muscular lipid stores (*Lie & Huse, 1992*; *Waagbø & Hansen, 1997*). This adaptation can be observed as reduced daily loss of body mass over time and this loss seems to be temperature dependent. The daily body mass loss over a period of 28 days was higher for trout and carp reared at 20 °C than at 10 °C (*Petri, 2003*). *Salem et al. (2007)* reported that three weeks starvation of rainbow trout reduced liver expression of genes involved in aerobic respiration, blood functions and immune responses, associated with a decrease in tissue metabolism. Further, an overall reduction in protein synthetic capacity was observed, and impairment of mitochondrial (aerobic) ATP production, while maintaining liver glycolytic and gluconeogenic competence. In lipid metabolism, down-regulated expression in pathways associated with hepatic lipid and fatty acid transport were seen, while maintaining fatty acid oxidation mechanisms. Thus, fish may maintain tighter control on the mechanisms of protein metabolism than metabolism of lipid or carbohydrate under short term starvation.

In the present study, we aimed to examine the physiological response to starvation and whether two weeks of starvation affected the robustness to physical handling in adult Atlantic salmon farmed under practical large scale farming conditions at low temperatures (4−5 °C).

## MATERIALS AND METHODS

### Fish and sampling

The present starvation study was conducted with adult Atlantic salmon (*Salmo salar* L.) at the large scale R&D site Centre for Aquaculture Competence (CAC), located in Langavika in Gardsundfjorden, Hjelmeland (Western Norway) with approval from the Norwegian authorities (Directorate of Fisheries, approval # R-HM-20). The overall experiment and sampling were controlled by a veterinarian and conducted according to the Norwegian Animal Welfare Act. These studies did not require special approval from the authorities. Technical details of the site were previously described by *Waagbø et al. (2013)*.

The study included examination of required days of starvation for a complete gut evacuation. We examined selected clinical and gene expression markers from liver, muscle, heart and skin tissues during the two week starvation time. The outcome was related to plasma cortisol as a traditional primary response stress marker (*Wedemeyer, 1997*; *Iwama, Afonso & Vijayan, 2006*). At each sampling time, corresponding groups of fish were sampled either directly from the cages or following a 45 min period with practical relevant moderate confinement stress (crowding), to examine if starvation affected the short time homeostasis to stress.

A population of adult Atlantic salmon [body weight 5,608 ± 1,205 g (SD); length 73 ± 4 cm (SD) and condition factor 1.43 ± 0.12 (SD); $n = 40$ at start] had been reared in one of twelve 24 m × 24 m steel cages (last period 30 m deep) from sea transfer until harvest. At the start of the production in September 2012, the cage was stocked with 50 000 S0 smolts of approx. 80 g body weight. Fish were fed a standard extruded diet (Skretting, Stavanger) of appropriate pellet sizes according to the increasing fish size during the 18 months production. Details on feed and biological performance are reported elsewhere (*Sissener et al., 2016*). The water temperature at the period of starvation averaged 4 °C (5 m depth; range 3.8–4.2 °C). The fish were sampled at four time points after feeding was terminated; initially (March 6th 2014) and at days 3, 7 and 14 of starvation. At each of the sampling points, fish were either sampled immediately after careful netting (Netted) or after a 45 min crowding stress in a narrowed catching net, similar to natural handling during harvest operations (Stressed). Fifteen Netted fish were rapidly collected (3–5 fish at a time) by careful netting and immediately killed by a blow to the head, aiming at minimizing stress. Each fish was weighed and length measured. Blood was collected from the caudal vein and selected tissues (liver, heart and skin) dissected and conserved. For Stressed fish, 15 fish were sampled after 45 min crowding stress, killed by a blow to the head, blood was collected from the caudal vein and selected tissues (liver and heart) were sampled and conserved as for Netted fish. Both groups were examined for gut content.

At each sampling, additional 10 fish were sampled for examining of gut content, making a total of 40 fish. At additional samplings at days 1, 2, 4, 5 and 6, forty fish were only weighed, length measured and examined for gut content after removal of the intestine.

Blood was sampled on heparin vacutainers (BD Vacutainer, Boston US) and kept chilled until centrifugation to collect plasma which was immediately frozen. Hearts were sampled by dissecting out a piece of the ventricle apex stored on RNAlater (Sigma-Aldrich, MO,

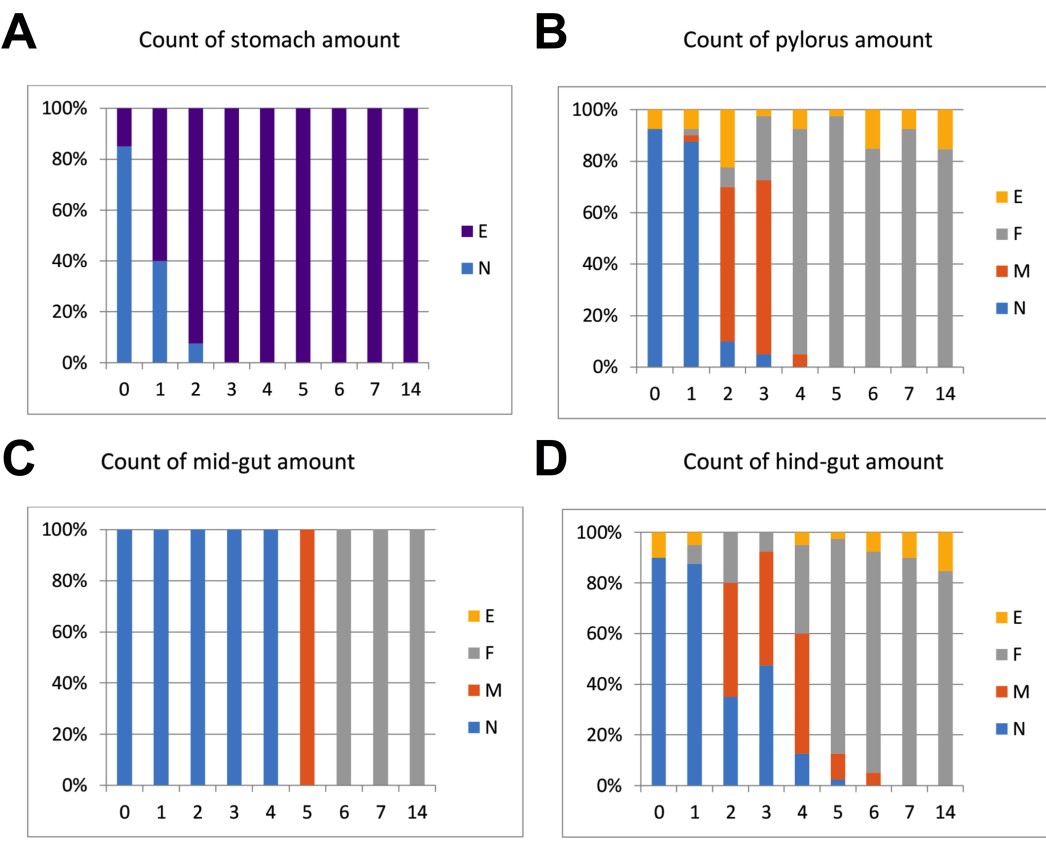

**Figure 1** **%-Distribution of in fish per sampling day (*n* = 40 fish) with content in stomach, pylorus, mid- and hind-gut.** The content was categorized as N (normal), F (flocculants), M (mix/intermediate between normal and flocculants) or E (empty). Flocculants was regarded as normal remnants of faeces following several days of starvation, not negatively affecting slaughter hygiene. The results in the graphs suggest that Atlantic salmon should be starved at least 5–7 days at low temperatures (4 °C) to empty the intestine.

US). Further, a piece of the liver was stored on RNAlater. A piece of skin and muscle was sampled from the NQC region (Norwegian Quality Cut - part of the salmon behind the dorsal fin, defined as reference fillet for quality measures) from the same side and stored on RNAlater.

## Measurement of gut content

The intestinal duct was carefully opened from oesophagus to hind-gut. The content of stomach, pylorus, mid- and hind-gut was characterized separately, and the content was categorized as N (normal), F (flocculants), M (mix/intermediate between normal and flocculants) or E (empty). Flocculants were regarded as normal remnants of faeces following several days of starvation, not negatively affecting slaughter hygiene (Fig. 1). The gut evacuation time was evaluated as the sampling day (0 to 7) where no gut content was observed, except remnants defined as "flocculants".

## Clinical analyses in plasma and bile

Plasma cortisol, glucose, lysozyme, protein and osmolarity were analysed from 15 Netted and 15 Stressed fish at day 0, 7 and 14 after starvation. Sampled bile from Netted fish only was analysed for cortisol and osmolarity at day 0 and 14 ($n = 12$–15). Plasma and bile were analysed for cortisol using a commercially available radioimmunoassay (RIA) kit, GammaCoat[TM] Cortisol [125]I radio immune assay kit (DiaSorin CA1529E, Saluggia, Italy). Plasma glucose, lysozyme and protein were analysed using a clinical bioanalyzer (Maxmat PL analyzer, Montpellier, France) according to standardized procedures, reagents and controls. Osmolarity was analysed in plasma and bile by measuring the freezing point with Fiske One Ten Osmometer (Fiske Associated, Norwood, MA, US).

## Gene expression of metabolic markers in liver

Table 1 shows selected metabolic gene markers (PCR primers and efficiencies) for starvation and stress within energy metabolism, oxidative health and overall protein turnover in liver tissue. Expression of these genes were analysed from 15 fish from day 0 and 14 post starvation, and from Netted and Stressed groups. Total RNA from liver was extracted using the BioRobot EZ1 and RNA Tissue Mini Kit (QiagenAB, Sollentuna, Sweden). Reverse transcription was performed using Multiscribe reverse transcriptase (Applied Biosystems, Foster City, California US). Real-time PCR was performed using SYBR Green Master Mix and the LightCycler 480 Real-Time PCR System (Roche Applied Sciences, Penzberg, Germany). Mean normalized expression (MNE) of the target genes was determined using a normalization factor based upon three un-regulated reference genes (ß-actin, Elongation factor 1 alpha B and ubiquitin-60S ribosomal protein L40; Table 1), as calculated by the *geNorm* software (*Vandesompele et al., 2002*).

## Gene expression of stress and health markers in heart and skin

Table 1 shows the selected stress signalling, inflammation and muscle contraction gene markers (PCR primers and efficiencies) analysed in heart and skin tissues. Heart samples were analysed, with 15 fish per time point (0, 3, 7, 14 days starvation) for the Netted group, and 15 fish per time point (0 and 14 starvation) for the Stressed group. Skin samples were analysed in 15 fish at start and after 3, 7 and 14 days starvation from the Netted group only. Total RNA from heart and skin tissues was extracted from frozen tissues and cDNA synthetized with SuperScript VILO kit (Applied Biosystems) using standard operational procedures. The reference gene EF1A was used for normalisation of gene expression in heart and skin tissues. Ct values were calculated by 2nd derivative max method as part of the LightCycler software. Relative expression was calculated according to the Pfaffl method (*Pfaffl, 2001*) adjusted for PCR efficiency.

## Statistics

The somatic data are given as mean (SD) while the other data are given as mean (SE or pooled SE). The treatment groups were compared with two-way ANOVA (Starvation, Crowding stress and their interaction term) and Tukey's post hoc test, and graphs prepared in Graphpad Prism. Skin analyses in Netted fish only was analysed by one-way ANOVA. In case of unequal variances, as determined by Bartlett's test, log-transformed data were

Waagbø et al. (2017), *PeerJ*, DOI 10.7717/peerj.3273

**Table 1** Tissue gene markers used in the experiment, their functional role and expected response, as well as their detailed characteristics.

| Gene | Functional role | Expected response | Accession number | Forward primer | Reverse primer | Amplicon size | PCR efficiency |
|---|---|---|---|---|---|---|---|
| **Liver tissue** | **Oxidative stress, cellular stress** | | | | | | |
| Mn SOD | Manganese superoxide dismutase | Down regulated | DY718412 | ATCACAGCCTGCCCTAATCAA | CGTAGTCGGGTCGGACATTC | 121 | 1.84 |
| HSP70 | Heat shock protein 70 | Down regulated | C169R048 | GTGCAGGCTGCCATCTTAGC | CATGACCCCTCCAGCTGTCT | 108 | 2.06 |
| | **Energy metabolism** | | | | | | |
| IGFBP1B | Insulin-like growth factor binding protein 1B | Up regulated | AY662657 | GAGGACCAGGGACAAGAGAAAGT | GCACCCTCATTTTTGGTGTCA | 128 | 1.83 |
| | **Stress response** | | | | | | |
| GRP78 | 78 kDa glucose-regulated protein precursor (GRP 78) | Up regulated | AM042306 | CCCCAGATCGAGGTCACCTT | TCCTCAGGCGTCAGACGATT | 128 | 2.01 |
| | **Lipid and steroid metabolism** | | | | | | |
| HMGCR | 3-hydroxy-3-methyl-glutaryl-coenzymeA reductase gene | Up regulated | Contig1955_Atlantic_salmon | GGTCCTGTGATTAGGTGCCC | AAACCTGCTGGTGTGGTCAA | 114 | 1.98 |
| ACTB | Beta-actin | Reference gene | BG933897 | CCAAAGCCAACAGGGAGAA | AGGGACAACACTGCCTGGAT | 91 | 1.88 |
| EF1AB | Elongation factor 1 alpha B | Reference gene | BG933853 | TGCCCCTCCAGGATGTCTAC | CACGGCCCACAGGTACTG | 59 | 2.00 |
| UBA52 | ubiquitin-60S ribosomal protein L40 | Reference gene | GO050814 | TCAAGGCCAAGATCCAGGAT | CGCAGCACAAGATGCAGAGT | 139 | 2.01 |
| **Heart and skin tissues** | **Stress signaling** | | | | | | |
| IERG2 | Immediate early response gene 2 | | NM_001140121.1 | CCAAGCGTTAACCGAACAGT | CTCGGGAGGCGTACAGTTTA | 122 | 1.87 |
| JUNB | Jun B proto-oncogene | | NM_001139901.1 | CTATCGGAACCAAAGCCTCA | GGATGATCAATCGCTCCAGT | 122 | 1.83 |
| | **Inflammation & muscle contraction** | | | | | | |
| TNF1A | Tumor necrosis factor 1 alpha | | DQ787157.1 | AGGTTGGCTATGGAGGCTGT | TCTGCTTCAATGTATGGTGGG | 400 | 1.73 |
| INOS | Inducible nitric oxide synthase | | AF088999.1 | ACAGACATTGGCCCAGAGAC | CTCCATTCCCAAAGGTGCTA | 140 | 2.01 |
| RYR1 | Ryanodine receptor isoform 1 | | DW541352 | CTCTACCGGGTGGTCTTTGA | ACCTGCTCTTGTTGGTCTCG | 119 | 1.97 |
| | **Mucosal & epithelial integrity** | | | | | | |
| MUC5 | Mucin-5B-like | | XM_014188489.1 | TACCAGGAGCCAGGCAGTTG | GCTGCACTGCTTCTGTGACC | 145 | 2.02 |
| MMP9 | Matrix metalloproteinase-9 | | NM_001140457.1 | CTGGCGCAGATATTTTGGAT | CATGGCTTTTGAGCCAGTTC | 133 | 2.04 |
| EF1A | Elongation factor 1 alpha | Reference gene | NM_001123629.1 | CACCACCGGCCATCTGATCTACAA | TCAGCAGCCTCCTTCTCGAACTTC | 78 | 1.91 |

**Table 2** Plasma clinical analyses of adult Atlantic salmon during after 7 and 14 days starvation (Netted), including a short-time confinement stress (Stressed) at each time point.

| Treatment | Day | Cortisol ng/mL | Glucose mM | Lysozyme U/g | Protein g/L | Osmolarity mOsm/L | n |
|---|---|---|---|---|---|---|---|
| Netted | 0 | 48 | 5.21 | 0.32 | 32.1 | 332 | 15 |
| Stressed | 0 | 77 | 5.45 | 0.29 | 36.8 | 344 | 15 |
| Netted | 7 | 83 | 4.35 | 0.40 | 37.4 | 325 | 15 |
| Stressed | 7 | 112 | 5.44 | 0.36 | 37.9 | 342 | 15 |
| Netted | 14 | 49 | 4.39 | 0.32 | 37.7 | 328 | 15 |
| Stressed | 14 | 90 | 4.99 | 0.35 | 39.4 | 342 | 15 |
| Pooled SEM | | 4 | 0.10 | 0.01 | 0.7 | 2 | |
| *Two-way ANOVA* | | | | | | | |
| Starvation | | $p < 0.005$ | $p < 0.018$ | ns | $p < 0.035$ | ns | |
| Stress | | $p < 0.001$ | $p < 0.001$ | ns | ns | $p < 0.001$ | |
| Interaction term | | ns | ns | ns | ns | ns | |

used for the ANOVA (for liver MnSOD, HSP70 and HMGCR; heart IERG2, JUNB, TNF1A and INOS, and skin MUC5 and MMP9 gene expressions). Individual correlation analysis between the parameters was done by a Spearman rank order correlation test (significant at $p < 0.05$).

## RESULTS

### Somatic data and gut evacuation time

Fish weight [$5347 \pm 940$ g (SD), $n = 360$], length [$72.0 \pm 3.9$ cm (SD), $n = 360$] and condition factor [$1.42 \pm 0.11$ (SD), $n = 360$] were similar for all sampled fish during the 14 day study, with only a marginal increase in length from $72.8 \pm 3.9$ (SD) to $74.4 \pm 3.2$ (SD) cm ($n = 40$ fish, $p < 0.047$), comparing the start and final samplings.

Regarding gut evacuation rates at low temperatures, Fig. 1 shows % of fish ($n = 40$ per day) with gut content in four sections (stomach, pylorus, mid-gut and hind-gut) during 14 days. The sections are gradually emptied with time, with 3, 5, 6 and 7 days in stomach, pylorus, mid-gut and hind-gut, respectively. Overall, the results showed that faeces was still found in gut sections from fish sampled on days 5 and 6 (mix/intermediate between normal and flocculants), implying that 7 days of starvation was needed to completely empty the gut at low temperature ($4-5\,°C$).

### Plasma clinical and bile analyses

The plasma cortisol values (Table 2) were generally low and indicated a moderately transiently increased concentrations with starvation ($p < 0.005$) and increased levels after confinement stress ($p < 0.001$). Plasma cortisol increased temporarily from 62 to 97 ng/mL after one week starvation, and returned to 69 ng/mL after 14 days (average levels of Netted and Stressed groups).

From Table 2, plasma protein increased significantly with starvation time ($p < 0.05$), but was not affected by confinement stress. Plasma osmolarity showed no changes with starvation, while an approx. 5% rise in concentration was observed in Stressed versus

Netted fish in all samplings ($p < 0.001$). For the entire sampled material, individual plasma cortisol was significantly positively ($p < 0.05$) related to plasma protein (Spearman rank order correlation $r = 0.39$; $n = 45$) and plasma osmolarity (Spearman rank order correlation $r = 0.43$; $n = 45$). Plasma glucose declined moderately but significantly with time of starvation (from 5.3 to 4.7 mM; $p < 0.02$), and increased secondary to stress (4.6 to 5.3 mM; $p < 0.001$). Plasma lysozyme activity was neither affected by starvation nor stress (Table 2). None of the clinical markers showed significant interaction terms between starvation and stress.

## Bile cortisol and osmolarity

Bile cortisol and osmolarity were analysed from the Netted samplings at start and after 14 days starvation, meaning that starvation time was the only experimental variable. Despite that mean cortisol values varied considerably at both time points, with $2{,}210 \pm 710$ (SE) vs $350 \pm 610$ (SE) ng/mL, individual bile cortisol concentration was significantly related to resting plasma cortisol (Spearman rank order correlation 0.56; $n = 27$). Bile cortisol was also related to bile osmolarity (Spearman rank order correlation 0.76). Bile osmolarity declined significantly from $430 \pm 21$ (SE) to $323 \pm 18$ (SE) mOsmol/L during starvation.

## Expression of metabolic gene markers in liver

Gene expression markers of the liver were chosen to reflect changes in energy metabolism (lipid and amino acid metabolism), oxidative status and overall protein turnover (Table 1), and how the fish copes metabolically to acute stress at start and following 14 days of starvation. Significant effects were observed after two weeks starvation on all genes (Fig. 2), validating the usefulness of the chosen genes in starvation with respect to oxidation (down-regulated MnSOD; $p = 0.0118$), cellular stress (up-regulated HSP70; $p = 0.0001$), growth and energy metabolism (down-regulated IGFBP1B; $p = 0.0006$), nutritional stress (down-regulated GRP78; $p = 0.0001$), and lipid and steroid metabolism (down-regulated HMGCR; $p = 0.0001$). In the present study, ubiquitin (uba52) was included as reference gene, indicating that starvation or stress did not have any major impact on protein degradation.

No significant effect was seen in the present gene markers in confined fish (Stressed) versus gently Netted fish for any of the examined genes ($p > 0.05$), including the HMGCR gene (Fig. 2). This latter gene was chosen to explore possible differences in cortisol synthesis from cholesterol.

## Expression of stress signalling and inflammation gene markers in heart

Figure 3 presents expression of two early stress signalling markers (IERG2 and JUNB) in heart tissue from Netted and Stressed fish over time after starvation. Both genes show similar significant temporal changes in expression pattern during starvation ($p = 0.0066$ and $p = 0.0006$, respectively). In the Netted group, both expression levels declined temporarily from 0 to 7 day of starvation, and then returning to initial levels after 14 days starvation. In the Stressed group, expression levels of both genes were generally significantly higher compared with the Netted group, confirming their immediate response to crowding stress

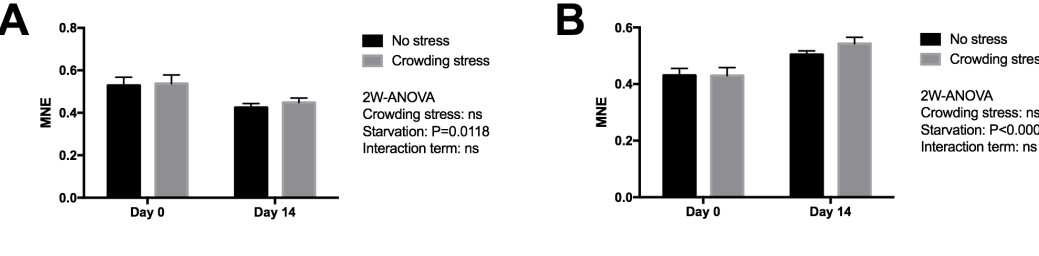

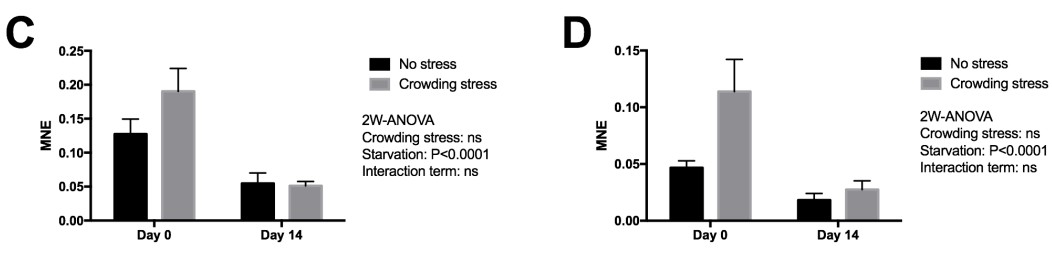

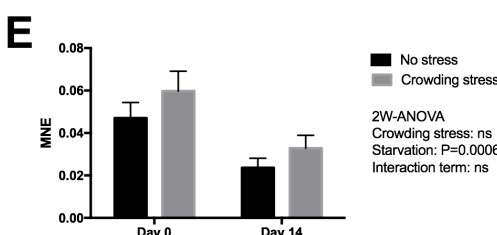

**Figure 2** Gene expression of proteins in liver showed that all markers ((A) MnSOD, (B) HSP70, (C) GRP78, (D) HMGCR and (E) IGFBP1 genes) responded to starvation, and none to confinement stress (the MnSOD, HSP70 and HMGCR data were log transformed). Two-way ANOVA was used to search for effects of crowding and starvation between Day 0 and Day 14. Significance levels of crowding stress, starvation and interaction terms are given. ns, not significant; MNE, mean normalized expression.

(both $p < 0.0001$). Expression of the cytokine TNF1a, a mediator of inflammation, cell survival and differentiation, showed higher levels in Stressed versus Netted fish ($p = 0.0431$; Fig. 3). Within the groups there were no changes in expression levels over the two weeks starvation. Expression levels of *inos*, a marker of cardiovascular function and inflammation, was affected by stress but not starvation (Fig. 3), with reduced expression observed in Netted fish ($p = 0.029$). Expression levels of *ryr1*, involved in muscle contractility through regulation of myofiber contraction, showed neither changes with starvation nor after confinement stress (Fig. 3).

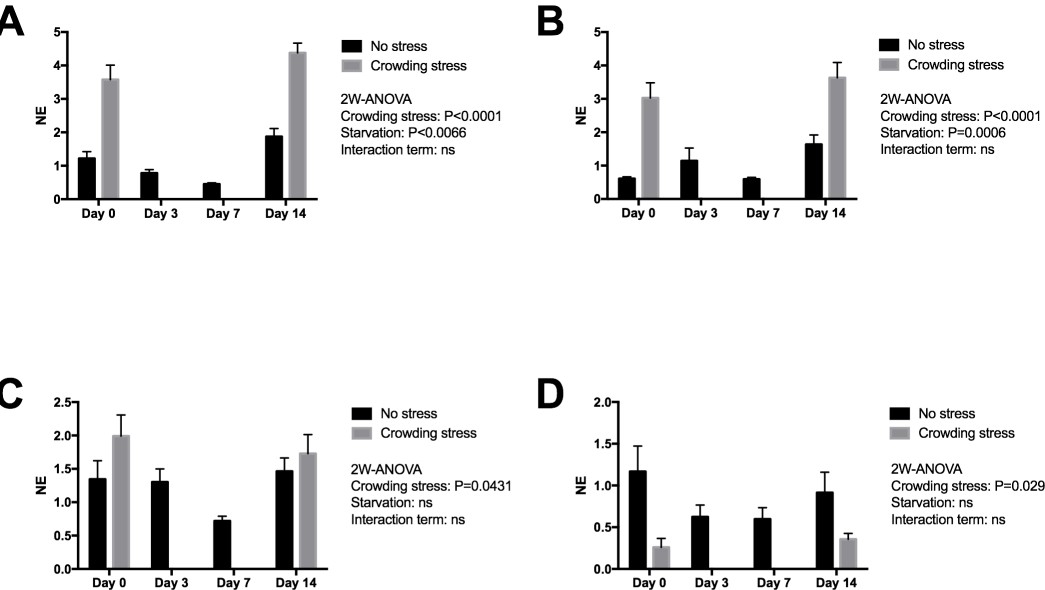

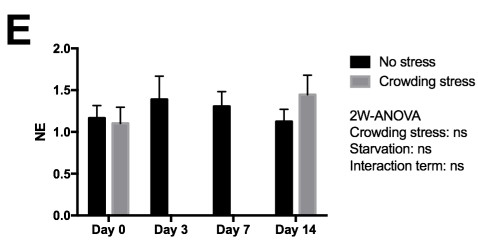

**Figure 3** Gene expression of proteins in heart tissue ((A) IERG2, (B) JUNB, (C) TNF1A, (D) INOS and (E) RYR1) showed that the selected marker IERG2 and JUNB responded differently to starvation, while the IERG2, JUNB, TNF1A (log transformed) and INOS (log transformed) genes responded significantly to confinement stress. Two-way ANOVA was used to search for effects of crowding and starvation between Day 0 and Day 14. Significance levels of crowding stress, starvation and interaction terms are given. ns, not significant; NE, Normalized expression.

## Expression of mucosal and epithelial integrity gene markers in skin

Skin tissue gene expression of mucin 5 (MUC5) and matrix metalloproteinase 9 (MMP9) were assessed in Netted fish only, and showed minor non-significant declines from day 3 to 14 of starvation (Fig. 4), with lowest expression levels after 14 days of starvation.

## DISCUSSION

The objectives of this study were to examine welfare aspects, including the response to acute stress, of two weeks starvation at low temperatures in adult Atlantic salmon prior to harvest. According to the quality legislation, the fish should be starved so no feed remnants are found in the stomach or intestines. The present study examined the gut evacuation time

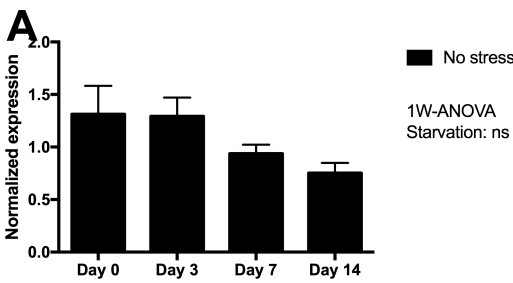

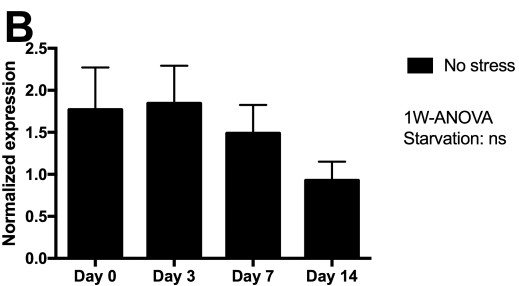

**Figure 4** Gene expression of proteins in skin tissue from Netted fish (No stress) showed that neither (A) MUC5 nor (B) MMP9 genes (log transformed) responded significantly to starvation. ns, not significant.

for fourteen days starvation at low temperature, and underway the fish were examined for physiological, metabolic and welfare issues, including the short-term response to confinement stress. The frame of metabolic stress by starvation and acute confinement stress was measured by plasma and bile cortisol concentrations, related plasma clinical markers, and the regulation of a number of relevant genes in liver, heart and skin tissues.

## Gut evacuation time

The present study demonstrates that current common practices with 3–4 days starvation prior to harvest (*Rasmussen, 2001*; *Einen, Waagan & Thomassen, 1998*) is not sufficient for complete removal of feed from the stomach and gut under the low temperature conditions of adult Atlantic salmon. The results suggest that fish should be starved for at least 5–7 days at such low temperatures. This is according to *Usher, Talbot & Eddy (1991)*, where calculated evacuation time was found two times longer at 4 °C than at 13 °C. Earlier studies have also shown that the intestinal evacuation time was inversely related to size of the previous feed intake in starved juvenile salmon (*Talbot, Higgins & Shanks, 1984*). In adult commercially grown salmon like in the present study, the fish had an optimal feed intake prior to starvation and the measured feed evacuation time is therefore representative for present Atlantic salmon farming at cold temperature.

## Stress responses

Plasma cortisol values were transiently and moderately increased at 7 days starvation, after which it returned to initial levels after 2 weeks in non-stressed fish, and the relative rises after 45 min confinement stress did not seem to be influenced by starvation time.

While plasma cortisol is a classical primary response marker to physical stress (*Wendelaar Bonga, 1997*), evidences of changes in glucocorticoids in response to fasting in fish are contradictory. Plasma cortisol levels in otherwise unstressed fish are variously reported to be unaffected, reduced or increased by fasting. Starvation implies changes in circulating hormones and expression of their receptors in target tissues, i.e., mainly affecting organs that take part in energy metabolism. *Pottinger, Rand-Weaver & Sumpter (2003)* found no significant changes in plasma cortisol levels at any point during long-term fasting of rainbow trout. Their results suggested that energy mobilisation during fasting may be achieved without the endocrine involvement of growth hormone, cortisol or somatolactin in line with results of an earlier study (*Kakizawa et al., 1995*). Similarly, Rosten et al. (cited in *VKM, 2008*) found that plasma cortisol recorded regularly during 15 hrs transportation was generally lower in salmon parr starved for 6 days than in fish starved for 2 days prior to transportation. *Mørkøre et al. (2008)* indicated improved resistance to acute stress in long-term starved fish (35 days), especially for *post mortem* quality aspects. The mechanisms for this was, however not clear and the response to stress was not confirmed by plasma cortisol analyses. In a marine species like gilthead seabream (*Sparus auratus*) exposed to 14 days starvation, however, several fold increase in plasma cortisol was observed (*Polakof et al., 2006*), which was attributed to increased carbohydrate (gluconeogenesis) and amino acid (transamination) metabolism to support energy. Similarly, *Costas et al. (2012)* found elevated cortisol and amino acids in plasma of Senegalese sole (*Solea senegalensis* Kaup, 1858) starved for 21 days and suggested a functional role of cortisol in energy mobilisation. There is obviously species, age, and conditional differences in the endocrine regulation of metabolism during starvation, where well-conditioned adult Atlantic salmon reared at low temperatures are less dependent on cortisol induced energy mobilisation.

The present experiment included analyses of bile cortisol. It is assumed that bile, in line with hair cortisol analysis in humans (*Wikenius et al., 2016*), might reflect longer-term and chronic stress posed by for example starvation (*Pottinger, 2008*). The decline in bile cortisol and osmolarity after 14 days of starvation reflected the severely reduced intestinal activity and digestion after longer food deprivation. In line with plasma cortisol, the concentration of bile cortisol decreased after 14 days starvation and there was a positive individual correlation between the two in starved non-stressed fish. However, large variation in the bile cortisol concentration reflected both inhomogeneous mucoid bile samples and the changes in composition with time of starvation with reduced bile production and volumes of bile in the gall bladder. The analysis of bile cortisol may therefore be useful to assess changes in cortisol over time, like the endocrine elevation during parr-smolt transformation in salmonids (*Shrimpton, Bernier & Randall, 1994*) and in periods of chronic stress. Since the production and storage of bile varies and declines under starvation, bile cortisol may be more successfully used as an indicator for chronically stressed fish under normal feeding regimes.

Similarly to plasma and bile cortisol concentrations, the liver HMGCR gene expression, representing cholesterol and cortisol synthesis was reduced with starvation. At start of the feed deprivation, the short-term stress seemed to activate the HMGCR gene (not

significant in the applied two-way ANOVA model), while it was not responsive after 2 weeks of starvation.

Plasma osmolarity and protein were chosen as markers to examine secondary stress induced physiological disturbances in the hydro mineral balance (increased osmolarity) and protein mobilization during fasting, respectively. The moderate increase in plasma protein with time seemed therefore to reflect a normal starvation metabolism, with mobilization of labile protein reserves for energy purposes. Although lower than given reference values of plasma protein in adult Atlantic salmon, the present rise was less than the seasonal variation (*Sandnes, Lie & Waagbø, 1988*). The use of ubiquitin as reference gene supports the fact that starvation or stress did not have any major impact on liver protein degradation. This is in contrast to 400 g gilthead seabream, where moderate declines in plasma protein were observed in groups starved for 14 days, concomitantly to elevated liver transaminase activity, supporting breakdown of amino acids for energy production (*Polakof et al., 2006*). Besides species differences, the discrepancy may well be related to rearing temperature, as seen for plasma cortisol and amino acid metabolism in Senegalese sole reared at cold and warm temperatures (*Costas et al., 2012*).

Plasma osmolarity was not affected by 14 days starvation, while it was increased after acute stress in all samplings. This is in line with observations in gilthead seabream starved for the same period (*Polakof et al., 2006*) and reflects the priority of homeostasis. Moderate decline in blood glucose was seen with starvation, while hyperglycemia is a classical secondary response confirming glucose mobilisation following stress and cortisol release (*Pottinger, 2008*). Together with rise in plasma cortisol, the secondary moderately elevated osmolarity and hyperglycemia in the confined fish, the clinical markers together demonstrated a classical mild response to stress in the present experiment. In a recent study where 2 kg Atlantic salmon exposed to high seawater temperature responded by reduced feed intake and anorexia, there was no impact observed on clinical parameters or liver and white muscle fatty acid composition during a period of 8 weeks (*Hevrøy et al., 2010*). The fish showed reduced metabolism, however without any obvious physiological challenges. This is also in line with a previous study on adult salmon during fasting (*Waagbø & Hansen, 1997*). Thus, the present study was suitable to further explore the impact of pre-harvest starvation and responses to the physical acute stress on metabolic adaptations and fish health and immunity at low temperature, as examined at gene expression level in liver, heart and skin tissue.

## Metabolic responses

When fish are starved, energy-saving strategies are elicited to maintain the supply of nutrients to selected tissues, especially to the brain (*Soengas et al., 1996*). Liver is a central organ for nutrient channelling during starvation, both through accumulated nutrients and as a metabolic centre. This is observed by changes in metabolism, both by slowing down energy spenditure and relative changes among the energy substrates at starvation. For example, enzymes involved in lipid breakdown and protein degradation and turnover will generally be up-regulated during starvation, and lipid anabolic enzymes will be down-regulated (*Bauer et al., 2004*; *Costas et al., 2011*; *Jagoe et al., 2002*; *Lange et al., 2004*; *Polakof*
*et al., 2006*; *Salem et al., 2007*; *Suzuki et al., 2002*). Starvation includes liver responses on transcriptional level for genes related to oxidative stress, autophagy, energy metabolism, stress response, lipid and steroid metabolism and protein degradation (*Martin et al., 2010*; *Antonopoulou et al., 2013*; *Morales et al., 2004*; *Salem et al., 2007*). The liver expression markers analysed in the present fish material was selected to give information on how the fish prioritised energy distribution and coped metabolically following 14 days of fasting and after exposed to stress. Significant effects were observed after two weeks starvation on all genes (Fig. 2), validating the outcome with respect to redox defence (down regulated MnSOD), cellular stress (upregulated HSP70), growth and energy metabolism (down regulated IGFBP1B), nutritional stress (down regulated GRP78) and lipid and steroid metabolism (down regulated HMGCR). All the nutritional related gene markers were down-regulated, and indicated classical starvation responses like reduced metabolism, reduced oxidative pressure and sparing of nutrients, including reduced igfbp1b expression, reflecting change in catabolism. Circulating IGF-I has been regarded as an index of recent growth in fish under changing nutritional conditions (*Beckman, 2011*). According to *Shimizu et al. (2005)*, the binding proteins of IGF-I have important roles in regulating the metabolic actions of circulating IGF-I. In a later paper, *Shimizu et al. (2006)* showed that the Chinook salmon circulating IGFBP-1, increased during catabolic states such as fasting and stress. Results from a short-term starvation study (14 days) with Atlantic salmon also showed that the liver *igfbp1b* mRNA and protein levels of the corresponding circulating IGFBP increased during 14 days of starvation (*Hevrøy et al., 2010*). The discrepancy to the declined expression in present study may rely on both the initial nutritional status of the fish and ambient temperature, where the present study was conducted with pre-harvest salmon of good nutritionally condition at low temperature, as compared to 128 g postsmolt salmon and 10 °C in the study by *Hevrøy et al. (2010)*. The present study could not detect any weight reduction during the 14 days starvation. The maintenance of energy homeostasis during food deprivation in fish is directly related to the capacity for mobilization of energy reserves such as lipids and hepatic glycogen, at least during the initial stages of fasting, and depends on subsequent activation of hepatic gluconeogenesis and reduction in the rate of glucose utilization (*Sheridan & Mommsen, 1991*; *Navarro & Gutiérrez, 1995*). The high body fat stores in the present pre-harvest salmon would ensure capacity to endure starvation, while continuous use of labile protein reserves may impact immunity, as shown for selected functional immunological systems in adult salmon under far longer starvation periods (*Waagbø & Hansen, 1997*; *Waagbø, 2006*). This may ultimately end in compromised immunity, increased susceptibility to diseases and mortalities in less robust fish.

The moderately increased *hsp70* expression with 14 days fasting was observed, probably as a protective measure corresponding to general gene down regulations. HSP70 has, however been used with mixed success to reflect unfavourable farming conditions, like feed deprivation and handling stress (*Zarate & Bradley, 2003*; *Olsvik et al., 2011*). *Zarate & Bradley (2003)* examined how HSPs responded to hatchery stress in the Atlantic salmon and concluded that HSP70 is not a sensitive indicator to aquacultural disturbances like feed deprivation, anesthesia, capture stress, crowding stress, formalin, hyperoxia and hypoxia.
In accordance with *Olsvik, Lie & Hevrøy (2007)*, most of the examined genes in liver were not significantly regulated in confined fish versus gently netted fish, except for the trend of an upregulated *hmgcr* gene after short-term confinement stress (45 min) at start of the starvation period (not significant in the two-way ANOVA model). This gene was chosen to examine differences in steroid and cortisol synthesis from cholesterol after stress. Although the liver is not the major site for cortisol synthesis, *hmgcr* gene expression was modestly upregulated in line with the moderately elevated plasma cortisol. *Gornati et al. (2005)* showed that both the *hmgcr* and *hsp70* genes were upregulated in the liver of fish reared at high rearing densities. HMGCR therefore seems to be a useful early marker of the integrated stress response in Atlantic salmon, reflecting changes in steroid and lipid metabolism with both starvation and shortly after confinement stress.

The *igfbp1b* (both samplings) and *grp78* (start sampling) mRNA expressions indicated a trend towards increased expression after stress (not significant), as found in earlier studies (*Shimizu et al., 2011*). They also found that cortisol treatment induced both igfbp-1a gene expression and igfbp-1a protein in the blood, confirming their role in catabolic conditions like stress. Circulating IGFBP-1 is generally inhibitory to the IGF-1 action and the expression of *igfbp1b* is negatively correlated to individual growth rates is salmonids (*Kawaguchi et al., 2013*). For the weak rise in *igfbp1b* in the present study, one has to bear in mind that initiation of expression and the RNA turnover may vary between the genes, and that the present short time between the stress and sampling may have excluded genes as suitable stress markers in the present study. For example, *Olsvik et al. (2011)* clearly suggested that *hsp70* mRNA was a good indicator recorded after two days of handling stress in Atlantic salmon, while it was not affected in the present study, recorded 45 min after stress. A study by *Martin et al. (2010)* in salmon parr reported that 28 days of starvation and bacterial infection (furunculosis) had profound effects on the liver transcriptome, indicating that key components of the immune system were depressed during starvation. However, following infection the starved fish attempt to compensate for this immunosuppression by increasing expression of several key immune related genes to a greater extent than seen in fish fed prior to infection. The principle of being prepared for coming feeding event in periods of starvation have been seen in several fish species experiencing and successfully surviving longer periods of starvation.

## Health and immunity

Mounting an immune response requires energy and an increase in metabolic activity, and the effectiveness of the response may be related to body energy reserves. Plasma lysozyme is a simple and commonly used marker of innate immune competence in fish, as response to vaccination, infection and immunosuppression (*Waagbø, 2006*). In the present study, plasma lysozyme activity was related to protein concentration to prevent confounding effects of changes in water balance in stressed fish. In line with other immunological markers, plasma lysozyme values in our study did not indicate any impact of long time starvation and acute stress on immunity. This confirms the stability of innate immunity during short term fasting and the lack of priority of immunity at acute stress relative to physiologically regaining homeostasis.

The effect of starvation on cardiac stress and health markers was evaluated due to the integrated role of the heart in maintaining physiological and metabolic homeostasis in salmon. A previous study on mice showed a broad array of molecular events in response to starvation, related to lipid and glucose energy metabolism, signalling, cell structure and the immune system (*Suzuki et al., 2002*). The high on-growth of salmon during the last phase of the production cycle may affect cardiac health (e.g., induced epicarditis, remnants of virus-induced myopathies) possibly causing cardiorespiratory problems and risk of incidences following transport and other stressful pre-harvest events. Thus, it was relevant to assess whether prolonged starvation would affect myocardial function. The impact of starvation on cardiac stress was evaluated by two genes, *ierg2* and *junb*, previously identified as immediate markers of the primary stress response to diverse perturbations in salmon. The expression of *ierg2* showed a temporary decline, ending with higher than initial levels after 14 days starvation. Both genes were significantly upregulated in the Stressed versus Netted fish, confirming their response to confinement stress. The significantly reduced expression levels of both markers in Netted fish at day 7, returning to initial levels at day 14 suggests that fish exhibited a lower stress response compared to start and after 14 days. In response to crowding stress, there were no differences in expression levels and the immediate stress response in fish between 0 and 14 days of starvation. Plasma cortisol levels showed an opposite regulation of the heart stress markers, with rise in levels after 7 days returning to initial levels after 14 days of starvation. The opposite results between stress markers and cortisol is likely reflecting different kinetics of endocrine versus transcriptional regulation, or their involvement in different arms of the stress axis (immediate versus chronic stress). In terms of interpretation, transient short-term stress is a healthy response to regain homeostasis (allostasis), but long-term stress could indicate a negative impact on the fish health. Hence, the reduced cortisol levels following 14 days starvation in addition to unchanged regulation of immediate stress-marker suggest that starvation for two weeks has no impact on the stress response, and may in fact be more adaptive and beneficial than one week of starvation at low temperature.

Cardiac expression of the myokine TNF1a was induced in response to the crowding stress, however levels of this gene, as well as *inos*, were unchanged with starvation time, suggesting that prolonged starvation did not lead to elevated inflammatory levels in the heart. No changes in the expression level of *inos* with time from start to day 14, further indicates that long starvation is safe in this regard. One marker for myocardial function e.g., muscle contraction, the ryanodine receptor (*ryr1*) was neither affected by prolonged starvation. This gene has also been implicated in the regulation of ATP production in heart, thus steady state expression levels suggested that prolonged starvation and short-term stress did not have any negative effects on myocardial function.

The skin of fish is a dynamic tissue with cellular turnover known to be influenced by factors including stress and environmental conditions (*Iger, Balm & Wendelaar Bonga, 1994*). Maintenance of skin and epidermal integrity during starvation and stress is crucial for a proper physical and chemical barrier to challenging environmental conditions and salmon lice infestation (*Jensen, 2015*). Unchanged expression of mucin 5 (*muc5*) and matrix metalloproteinase 9 (*mmp9*) genes in starved fish suggests that prolonged starvation did

not have any negative effects on skin integrity in the present large scale experiment. A recent study on starvation of Atlantic salmon parr in freshwater showed that food deprivation for 18 days caused a rapid decrease in the densities of epidermal mucous cells, particularly in the lateral region of fish (*Landeira-Dabarca, Álvarez & Molist, 2014*). These changes may reflect reduced tissue turnover and activity during prolonged starvation. This is in line with previous observations in other fish species (*Caruso et al., 2010* in the eel (*Anguilla anguilla* L.) or *Somejo et al., 2004* in the Nile tilapia (*Oreochromis niloticus* L.) The trend towards reduced levels of skin *muc5* expression at days 7 and 14 could possibly imply a similar effect of prolonged starvation. The same trend was observed for *mmp9*, a gene that is typically upregulated in response to any perturbations of the skin tissue and extracellular environment.

## CONCLUSIONS

The data from the reported study imply that at low temperatures adult salmon with large accumulated body reserves prior to fasting may well handle starvation periods for two weeks, without any negative effects on general stress level, immunity or health, or on their ability to cope with acute physical stress. A mild and temporary rise in plasma cortisol after 7 days starvation was observed, while no change in primary and secondary responses to stress were seen after two weeks starvation. Down-regulation of all the selected nutritional related gene markers in liver indicated a classical response to starvation, like reduced metabolism, reduced oxidative pressure, increased cell protection and sparing of nutrients, including reduced igfbp1b expression reflecting catabolism. Starvation did not affect immunity, nor heart functions or skin integrity. The stress markers in heart indicated a predictable response to acute handling stress.

## ACKNOWLEDGEMENTS

The authors want to thank the technical staff at NIFES (Eva Mykkeltvedt, Jacob Wessels), Nofima, Ås and at CAC, Hjelmeland for their technical and analytical help.

### Funding

The research was funded by CAC (Hjelmeland, Norway). The CAC is a research and development site and an employed veterinarian researcher was involved in study design, sampling, data analysis and publication. The funders had no role in study design, data collection and analysis, decision to publish, or preparation of the manuscript.

### Grant Disclosures

The following grant information was disclosed by the authors:
CAC.

## Competing Interests

Dr. Olav Breck is an employee of Marine Harvest ASA. The authors declare there are no competing interests.

## Author Contributions

- Rune Waagbø, Sven Martin Jørgensen and Olav Breck conceived and designed the experiments, performed the experiments, analyzed the data, contributed reagents/materials/analysis tools, wrote the paper, prepared figures and/or tables, reviewed drafts of the paper.
- Gerrit Timmerhaus and Pål A. Olsvik performed the experiments, analyzed the data, contributed reagents/materials/analysis tools, wrote the paper, prepared figures and/or tables, reviewed drafts of the paper.

## Animal Ethics

The following information was supplied relating to ethical approvals (i.e., approving body and any reference numbers):

The data is achieved from a large scale R&D facility, CAC, Hjelmeland with approval from the Norwegian authorities (Directorate of Fisheries, Bergen, Norway; approval #R-HM-20).

## Data Availability

The raw data has been supplied as Supplementary Files.

## Supplemental Information

Supplemental information for this article can be found online at http://dx.doi.org/10.7717/peerj.3273#supplemental-information.

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
