# Peer review of "Short-term starvation at low temperature prior to harvest does not impact the health and acute stress response of adult Atlantic salmon"

_PeerJ, doi:10.7717/peerj.3273_

## Round 0.1 · original submission · Minor Revisions

Please revise the manuscript considering the 3 reviewers comments. Two of the reviewers have concerns on the Statistical analysis / description. I believe the experimental design based on sampling a population in a large cage is sound (your experimental unit is the individual fish, not the cage), and I do not share the concerns of reviewer 3 on the validity of ANOVA in this case. Nevertheless you should address in the revised version the other concerns of the two reviewers, including checking of ANOVA assumptions and providing the interaction term of the 2W-ANOVA.

·

Basic reporting

This Manuscript has good level of English.
I miss a little bit more the metabolic response in fish, more metabolic way. But the structural and information are well.

I need to see the efficiency of genes expression value en Mat-Method.

Experimental design

The manuscript describe original primary research and the hypothesis is clear, and method too.

The investigation was conducted rigorously and to a high technical standard.

The research was conducted in conformity with the prevailing ethical standards in the field.

Validity of the findings

The statistical method need to be clarify, because it necessary to test the Normality-Homocedasticity-Independence before to run Anova (one or two way).

Speculation is welcomed, but should be identified as such: here I would like more things about amino-acids, lactate or triglycerides routes. For example, Lactate levels in plasma why not was measured, may be that high levels of lactate or triglycerides are keeping the osmolality, what happend with GDH activity and amino-.acids way, because the liver decreased with starvation and fat also, but GDH need to moved the nitrogenous waste.

Additional comments

The manuscript is really interesting but has some things that need to be clarify (to see before) and changed as:
Osmolality = mOsm/kg
Osmolarity= mOsm/L
Cortisol is better to use ng/mL
SEM in table 2, and to put where appear statistical differences more than P-value.

In discussion
Please to include more information about Starvation-Fasting/stress/metabolic response, some example of articles:
Polakof et al 2006 J Comp Physiol B 176: 441–452,
Costas et al., Fish Physiol Biochem (2011) 37:495–504
Costas et al., Amino Acids (2012) 43:327–335

Reviewer 2 ·

Basic reporting

Please improve captions for each figure. It needs to be more descriptive. Please avoid acronyms.

Experimental design

The manuscript authored by Waagbø and colleagues is original and fits within the scope of the journal. Objectives clearly defined to explore the effects of starvation on the physiological response of adult Atlantic salmon, as well as to assess the implications of starvation (2 weeks) on some health indicators in salmon reared under practical large scale farming conditions at low temperature. The methods are sound and the research is in accordance to ethical standards.

Validity of the findings

No comments.

Additional comments

This is a very interesting paper worthy of publication however I do have a few comments that need addressing. TNF1a is indeed a pro-inflammatory cytokine. However, it belongs to a superfamily of pro-inflammatory cytokines which activate signalling pathways for cell survival, apoptosis, inflammatory responses, and cellular differentiation. Therefore, please change the assumption that TNF1a is an inflammatory marker by itself. In fact, it is actually estrange the authors are looking for inflammation according to the experimental design. Moreover, the increase in mRNA TNF1a levels is not supported by iNOS transcript data.

Reviewer 3 ·

Basic reporting

The work deals with a key factor in commercial aquaculture to ensure the fish health and welfare for harvesting. The writing is clear and relatively well structured though the big amount of variables and analysis make the reader lose the main objectives.
One of the major strengthens is the practical application of the experiment since the culture conditions were very similar to the commercial seafarms. Therefore the findings could be applied directly to the commercial seafarms.
However the statistical approach is not very appropriate and some assumptions for the application of some tests are not followed.
Finally, as said before, due to the big amount of data generated, the discussion is difficult to write and to understand, despite authors have divide it into several sections (I think it is a good method).

Experimental design

The main weakness of this experiment is probably the experimental design. The use of an only cage (experimental group) make the statistical methods to be applied are limited. For instance, one of the assumptions for an ANOVA is the independency between groups, which is not considered because all fish come from the same population/cage. Therefore, being statistically strict, other tests like related samples t-tests or Wilcoxon rank tests should be used. However it is known that this assumption is not considered in many works already published and there is no an alternative test for 2-way ANOVA in these cases. Moreover, the experiment and data collection are already done and those could not be changed to improve the manuscript.
Another questionable matter is the necessity of performing correlation tests. Firstly, I assume that data are not normal hence non-parametric test (Spearman) is used, this should be clarified in statistical section; else the suitable test would be Pearson correlation. Anyway, I think these correlations do not provide relevant information and mislead the reader; in fact, no comments about them are shown in the discussion.

Validity of the findings

In spite of the questionable statistical approach, the findings could be considered valid since that lack of statistical robustness is relatively common in many published findings which have been already verified later through other means. In other words, the assumption of dependency in ANOVAs is continuously violated and results do not seem to be different. Since the experiment was conducted under culture conditions similar to commercial facilities, the salmon culturists could adapt their methodologies to these findings.
As many analyses were performed, some of them were not considered in the discussion.
Although I like the way of presenting data in the discussion (divided into sections), I think those sections should be summarized and reduced; I suggest the next sections: Gut evacuation time; Stress responses; Metabolic responses; Health and immunity; Conclusions

Additional comments

Some particular comments are the next:
- L.91-100. This paragraph should be summarized, indicating the main objectives of the work, some sentences (i.e, from L.94) should be written in the M&M section.
- L112 and more. Please change the notation, use 0.00 ± 0.00 (SE or SD)
- L126. Please indicate which tissues were selected
- L144. Please join this section to fish and sampling section
- L252-253. This sentence should be placed into M&M section
- L227-235. Showing these data in a table would improve the data understanding
- Table 1: two-way ANOVA, the interaction coefficients are missing.

Although the text is understandable and clear, the authors should avoid showing non-relevant data and justify the necessity of performing all those analysis. I think some of them could be removed.
Overall, I think that the manuscript would be improved if these comments are taken into account and the discussion is adapted to those modifications.

---

## Round 0.2 · accepted · Accept

I was asked to step in to make the final Accept decision on your submission as the prior Editor is currently unavailable. I have read the revised manuscript and found it very interesting.

·

Basic reporting

All point are well done.

Experimental design

This point is correct and the authors follow the reviewer comments.

Validity of the findings

This point is correct and the authors follow the reviewer comments.

Additional comments

I'm agree with the authors revision and I do not have more comments. For me this manuscript is in condition to be accepted.

Reviewer 2 ·

Basic reporting

All comments have been addressed.

Experimental design

All comments have been addressed.

Validity of the findings

All comments have been addressed.

Additional comments

I am pleased to say you have satisfied my questions, thank you.

Reviewer 3 ·

Basic reporting

The authors have modified the manuscript according to reviewers' comments and clarified some items.
From my point of view, the current manuscript can be published in the journal.

Experimental design

No comment (see general comment)

Validity of the findings

No comment (see general comment)

Additional comments

No comment (see general comment)